# Analysis of chronic disease comorbidity patterns in middle-aged and elderly smokers in China: The China Health and Retirement Longitudinal Study

**Shanwen Sun[1], Yali Wang[1], Hailong Hou[2], Linlin Chen[1], Yuqi Niu[1], Yefan Shao[1], Xiaochun Zhang[1]***

1 The First Affiliated Hospital of China Medical University, Shenyang, China, 2 China Medical University School of Nursing, Shenyang, China

* xczhang73@cmu.edu.cn

## Abstract

### Background

China has the largest tobacco consumer population in the world, and with the increasing severity of aging, the health issues caused by smoking continue to draw attention. This study aims to explore the current state of chronic disease comorbidity under different smoking patterns, providing a foundation for the formulation of public health policies.

### Methods

A total of 10771 participants older than 45 were included from CHARLS collected in 2020 after screening. The research variables were extracted and processed using Stata 18.0 software, followed by data filtering and organization in Excel. Through chi-square tests and multinomial logistic regression analysis, the study explored the influencing factors of comorbidity patterns under different smoking statuses.

### Results

The results indicate that within the framework of multiple chronic conditions, 2,254 individuals (20.9%) were relatively healthy, while 79.1% had one or more chronic conditions: 3,656 in the simple chronic disease group (33.9%), 3,998 in the minor complex chronic disease group (37.1%), and 863 in the major complex chronic disease group (8.0%). The severity of comorbidity patterns was significantly associated with age, alcohol consumption, hospitalization in the past year, outpatient visits in the past month, insufficient sleep, and poor cognitive function across different smoking statuses. Additionally, living in urban areas and having depression were linked to higher comorbidity severity among non-smokers. Among smokers, being female and having low life satisfaction were independent risk factors for comorbidity patterns. In the group of former smokers, intense physical activity was associated with an increased risk of severe comorbidity patterns.

**Data availability statement:** Third party data for this study can be accessed from the China Health and Retirement Longitudinal Study website (http://charls.pku.edu.cn). The authors confirm that interested researchers would be able to access these data in the same manner as the authors. The authors also confirm that they had no special access privileges that others would not have.

**Funding:** The author(s) received no specific funding for this work.

**Competing interests:** The authors have declared that no competing interests exist.

**Abbreviations:** CHARLS, China Health and Retirement Longitudinal Study;CMS, Centers for Medicare & Medicaid Services;ETS, Environmental Tobacco Smoke;IPAQ, International Physical Activity Questionnaire;MCC, multiple chronic conditions;METs, metabolic equivalents;SB, Sedentary behavior.

## Conclusion

In response, it is recommended to strengthen public health strategies and interventions related to chronic disease comorbidity among smokers.

## Introduction

China's continuously growing economy has led to significant improvements in the quality of life for its citizens. However, it is crucial not to overlook the rapid shift in the country's demographic structure towards an aging population, which has led to the gradual emergence of social health issues. According to China's Healthy China 2030 planning outline, nearly 1.8 billion elderly individuals in China suffer from chronic diseases, accounting for 75% of the elderly population [1]. The prevalence of these conditions is steadily rising [2]. The National Health Commission's Report on Nutrition and Chronic Diseases of Chinese Residents (2020) indicates that comorbidity associated with chronic diseases greatly complicates both disease treatment and health management. The interplay between different types of chronic illnesses results in complex patterns of associative multimorbidity [3], which show significant regional variation.

Research indicates that individuals with comorbid chronic diseases experience both psychological and physical distress, severely impairing their quality of life. Yuan et al. points out that as the number of diseases an individual suffers from increases, there is a gradual emergence of disabilities, impairments, and mental stress [4], leading to a mortality rate several times higher compared to healthy individuals [5]. It is estimated that by 2030, the number of deaths from chronic diseases in China will reach 122 million, marking a 40.0% increase from 2013 [6]. On the other hand, the prolonged treatment period and frequent complications of chronic diseases impose a substantial economic burden on patients and their families. Data shows that medical expenses for patients with multiple chronic conditions(MCC) are nearly 5.5 times higher than those without multiple chronic conditions [7]. This undoubtedly represents a formidable social challenge. Many studies focus on the incidence and treatment of individual diseases, often overlooking the interactions between multiple diseases, which hampers the efficient utilization of health resources.

While age, gender, BMI, and family history are recognized as common demographic variables affecting multiple diseases, the impact of unhealthy lifestyles is also significant [5,8]. For instance, harmful habits such as smoking and excessive alcohol consumption escalate the risk of diseases and cause damage to multiple organ systems including the respiratory and cardiovascular systems [9]. China, being the largest producer and consumer of tobacco worldwide, harbors a vast smoking population [10]. During combustion, tobacco releases a complex mixture of chemicals that includes at least 86 known carcinogens and hundreds of toxic compounds. The inhalation of these harmful substances poses a serious threat to human health, contributing to the onset and progression of various chronic diseases. These compounds can damage cellular DNA, leading to genetic mutations that may trigger cancer [11]. Additionally, they damage the inner lining of blood vessels and accelerate the process of atherosclerosis, increasing the risk of cardiovascular diseases such as myocardial infarction and stroke. Smoking is also a major contributor to lung health problems, including pulmonary fibrosis, asthma, and chronic obstructive pulmonary disease (COPD) [12]. Moreover, smoking is linked to liver diseases, kidney diseases, and a range of autoimmune and inflammatory conditions. More concerning is that smoking may exacerbate the severity of these chronic diseases, increasing the difficulty and complexity of treatment, thereby having long-term and profound impacts on an individual's health [13]. The China Smoking and Health Report 2020 reveals that there

are over 300 million smokers in China. Tobacco use results in the loss of over one million lives annually, with projections rising to two million by 2030 and three million by 2050 [14]. This alarming situation highlights the dire and frightful connection between tobacco and health.

Previous studies on multiple chronic conditions, which simply define comorbidity as the presence of two or more chronic illnesses, may not accurately reflect the complexity and severity of comorbid chronic diseases, potentially reducing the reliability of findings [15]. Given the complex mechanisms of comorbidity, it becomes crucial to analyze the prevalence, patterns of illness, and potential related factors among the middle-aged and elderly. This study explores the current state of multiple chronic conditions under different smoking patterns, aiming to inform the development of public health policies and contribute to the continuous improvement of health policies and systems.

## Methods

### Participants

CHARLS utilizes a stratified, multistage, and population-proportionate probability sampling method to collect high-quality microdata on socioeconomic status and health conditions of households and individuals aged 45 and above, ensuring national representativeness [16]. The sample encompasses 450 villages, 150 counties, and 28 provinces, and has been widely used to study health-related issues among the elderly in China. This study focuses on middle-aged and elderly individuals aged 45 years and older. Given the ample sample size available and in consideration of ensuring data authenticity, this study employs a direct deletion approach for handling missing values, rather than imputing them through dummy variables. After excluding samples that did not meet the analysis criteria and those with missing key variables, the study included a total of 10,771 valid samples, as detailed in Fig 1.

The CHARLS surveys have received approval from the Biomedical Ethics Committee of Peking University, with the approval number IRB00001052-11015. Each respondent who agreed to participate in the survey was required to sign two copies of the informed consent

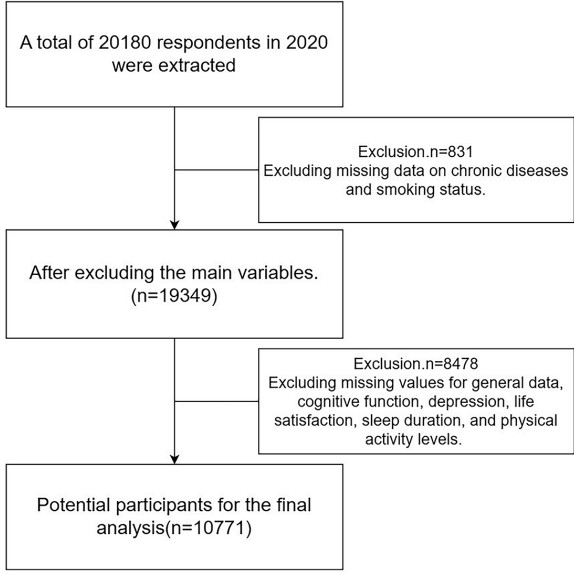

**Fig 1. Flow of participants in the study.**

form. One copy was kept by the respondent, and the other was stored in the CHARLS office and scanned in PDF format. All study participants were anonymized. This study is a secondary analysis of public databases, and ethical approval is not required for this research.

## Measures

In this study, the dependent variable is the pattern of multiple chronic conditions under different smoking statuses. The independent variables include demographic and related factors such as gender, age, marital status, educational level, place of residence, alcohol consumption, hospitalization in the past year, outpatient visits in the past month, level of physical activity, sleep duration, cognitive function, depression, and life satisfaction. Key variables among these are specified in detail to facilitate a deeper understanding of their impact on the comorbidity patterns observed in different smoking statuses.

## Definition of key independent variables

Physical activity levels are quantified using Metabolic Equivalents (METs), a metric for gauging the intensity of physical activities. According to the standards set by the International Physical Activity Questionnaire (IPAQ), the MET values are assigned as follows: walking is rated at 3.3 METs, moderate-intensity activities at 4.0 METs, and high-intensity activities at 8.0 METs. Based on this calculation, weekly physical activity levels are categorized into low (<600 METs/week), moderate (600–3000 METs/week), and high (>3000 METs/week) intensity [17].

Depression is measured using the CES-D scale, which consists of four levels with a composite scoring range from 0 to 30. Cognitive function is assessed through a survey encompassing 21 questions across four categories: everyday memory, word recall, arithmetic, and drawing. Each correct answer is scored as 1, and incorrect as 0, allowing for a comprehensive cognitive score ranging from 0 to 21 [18].

## Definition of the dependent variable

Based on the list of key chronic diseases by the Centers for Medicare & Medicaid Services (CMS) and definitions from related research, chronic diseases in this study are categorized into complex chronic conditions and non-complex chronic conditions. The study includes 14 types of somatic chronic diseases self-reported in CHARLS, comprising 8 complex chronic diseases (diabetes, chronic lung disease, heart disease, stroke, kidney disease, emotional/mental issues, dementia, and asthma) and 6 non-complex chronic diseases (hypertension, dyslipidemia, cancer, liver disease, gastrointestinal diseases, arthritis/rheumatism) [19,20].

Further, in accordance with established international research, the patterns of comorbidity among chronic somatic diseases are defined as follows: 1. Relatively Healthy Group: Individuals without any of the mentioned 14 chronic diseases. 2. Simple chronic illness: Individuals suffering only from 1-5 non-complex chronic diseases. 3. Minor complex chronic illness: Individuals with 1-2 complex chronic diseases and fewer than 6 non-complex chronic diseases. 4. Major complex chronic illness: Individuals with 3 or more complex chronic diseases or 6 or more non-complex chronic diseases [19].

## Statistical analysis

This paper utilizes data from the 2020 China Health and Retirement Longitudinal Study (CHARLS) national survey. The research variables were extracted and processed using Stata 18.0 software, followed by data filtering and organization in Excel. In practical data analysis, the listwise deletion method is commonly used to handle missing data, particularly when

the data are missing at random (MAR). This approach helps maintain the accuracy of the analysis results by excluding cases with missing values, ensuring that only complete data are analyzed [21]. The organized data were then imported into IBM SPSS Statistics 21.0 software for statistical analysis. Descriptive statistics, including mean ± standard deviation, frequency, and percentage, were used to analyze the general characteristics of the samples under different smoking statuses and the distribution patterns of comorbid chronic diseases. Univariate analysis was conducted using the chi-square ($\chi 2$) test.

In the multivariate analysis, since the number of levels of the dependent variable exceeds 2, multinomial logistic regression analysis is utilized to explore the influencing factors of comorbidity patterns under different smoking statuses, with a bilateral test level α set at 0.05.

## Results

### Basic characteristics of participants by smoking status

This study included a total of 10,771 participants. Overall, there were 6,507 non-smokers (60.4%), 2,873 current smokers (26.7%), and 1,391 former smokers (12.9%). The sample comprised 5,185 males (48.1%) and 5,586 females (51.9%), with an average age of 64.98 ± 9.44 years. Within the multiple chronic conditions framework, 2,254 individuals (20.9%) were relatively healthy, while 79.1% had one or more chronic conditions: 3,656 in the simple chronic disease group (33.9%), 3,998 in the minor complex chronic disease group (37.1%), and 863 in the major complex chronic disease group (8.0%), as shown in Table 1.

After categorizing by smoking status, univariate analyses were conducted on the patterns of multiple chronic conditions. Among non-smokers, statistically significant differences (P < 0.05) were observed in age, marital status, educational level, place of residence, alcohol consumption, hospitalization in the past year, outpatient visits in the past month, physical activity level, sleep duration, cognitive function, and depression. Among current smokers, significant differences (P < 0.05) were found in gender, age, marital status, educational level, alcohol consumption, hospitalization in the past year, outpatient visits in the past month, physical activity level, sleep duration, cognitive function, depression, and life satisfaction. Among former smokers, differences in gender, age, alcohol consumption, hospitalization in the past year, outpatient visits in the past month, physical activity level, sleep duration, and cognitive function were statistically significant (P < 0.05), as detailed in Table 2.

### Basic characteristics of diseases by comorbidity patterns

The 10,771 study participants were categorized into non-smokers, current smokers, and former smokers, and the prevalence of various diseases across different comorbidity patterns was described and compared. The results indicated that the top four prevalent diseases across different smoking statuses were hypertension, stomach/digestive system diseases, arthritis/rheumatism, and dyslipidemia. Unlike the non-smokers and former smokers, where heart disease ranked fifth, lung disease was the fifth most common among current smokers. Additionally, this study compared the disease prevalence across the three comorbidity patterns among the groups, revealing statistically significant differences in chronic diseases except for malignant tumors (P < 0.05), as shown in Table 3.

### Multivariate analysis of comorbidity patterns across different smoking statuses

Using comorbidity patterns among different smoking statuses as the dependent variable, factors statistically significant (P < 0.05) from univariate analysis were incorporated into a multinomial logistic regression. These factors include gender, age, marital status, education

**Table 1. Descriptive statistics of different smoking statuses.**

| Variable | Group | Smoking Status N(%)/ M±SD | | | Total |
|---|---|---|---|---|---|
| | | Non-smoker (n=6507) | Current Smoker (n=2873) | Former Smoker (n=1391) | |
| **Gender** | Male | 1255 (19.3) | 2657 (92.5) | 1273 (91.5) | 5185 (48.1) |
| | Female | 5252 (80.7) | 216 (7.5) | 118 (8.5) | 5586 (51.9) |
| **Age(years)** | | 64.49±9.52 | 64.67±9.05 | 67.90±9.30 | 64.98±9.44 |
| **Marital Status** | Married | 5440 (83.6) | 2507 (87.3) | 1225 (88.1) | 9172 (85.2) |
| | Other | 1067 (16.4) | 366 (12.7) | 166 (11.9) | 1599 (14.8) |
| **Education Level** | Primary School and Below | 4422 (68.0) | 1625 (56.6) | 780 (56.1) | 6827 (63.4) |
| | Middle School and Above | 2085 (32.0) | 1248 (43.4) | 611 (43.9) | 3944 (36.6) |
| **Place of Residence** | Rural | 2569 (39.5) | 1073 (37.3) | 594 (42.7) | 4236 (39.3) |
| | Urban | 3938 (60.5) | 1800 (62.7) | 797 (57.3) | 6535 (60.7) |
| **Alcohol Consumption** | No | 5012 (77.0) | 1153 (40.1) | 646 (46.4) | 6811 (63.2) |
| | Yes | 1495 (23.0) | 1720 (59.9) | 745 (53.6) | 3960 (36.8) |
| **Hospitalization** | No | 5352 (82.2) | 2435 (84.8) | 1007 (72.4) | 8794 (81.6) |
| | Yes | 1155 (17.8) | 438 (15.2) | 384 (27.6) | 1977 (18.4) |
| **Outpatient** | No | 5108 (78.5) | 2457 (85.5) | 1050 (75.5) | 8615 (80.0) |
| | Yes | 1399 (21.5) | 416 (14.5) | 341 (24.5) | 2156 (20.0) |
| **Physical Activity Level** | Mild | 964 (14.8) | 433 (15.1) | 230 (16.5) | 1627 (15.1) |
| | Moderate | 1680 (25.8) | 726 (25.3) | 437 (31.4) | 2843 (26.4) |
| | Severe | 3863 (59.4) | 1714 (59.7) | 724 (52.0) | 6301 (58.5) |
| **Sleep Duration(hours)** | | 5.99±1.88 | 6.25±1.77 | 6.12±1.77 | 6.07±1.84 |
| **Cognitive Function** | | 12.81±3.20 | 12.83±3.19 | 12.83±3.36 | 12.82±3.22 |
| **Depression** | | 7.79±6.22 | 8.03±6.15 | 8.16±6.39 | 8.01±6.23 |
| **Life Satisfaction** | | 3.27±0.75 | 3.25±0.73 | 3.28±0.74 | 3.27±0.74 |
| **Complex Chronic Disease Comorbidity Pattern** | Relatively Healthy Group | 1325 (20.4) | 746 (26.0) | 183 (13.2) | 2254 (20.9) |
| | Simple Chronic Disease Group | 2314 (35.6) | 963 (33.5) | 379 (27.2) | 3656 (33.9) |
| | Mildly Complex Chronic Disease Group | 2378 (36.5) | 977 (34.0) | 643 (46.2) | 3998 (37.1) |
| | Major Complex Chronic Disease Group | 490 (7.5) | 187 (6.5) | 186 (13.4) | 863 (8.0) |

level, place of residence, alcohol consumption, hospitalization in the past year, outpatient visits in the past month, physical activity level, sleep duration, cognitive function, depression, and life satisfaction, as shown in Table 4. The findings revealed:

Age, alcohol consumption, hospitalization in the past year, outpatient visits in the past month, sleep duration, and cognitive function were common influencing factors across the three smoking statuses. Specifically:With increasing age, the risk of more severe comorbid chronic disease patterns increased by 1.02–1.09 times. Individuals who consumed alcohol had a 20%-43% lower risk of severe comorbid chronic disease patterns. Patients who had outpatient visits in the past month faced a 2.22–11.02 times higher risk of severe comorbid chronic disease patterns. Patients who had been hospitalized in the past year had a 1.51–9.59 times higher risk of severe comorbid chronic disease patterns. An increase in sleep duration reduced the risk of severe comorbid chronic disease patterns by 10%-23%. Improvements in cognitive function reduced the risk of severe comorbid chronic disease patterns by 3%-9%.

Place of residence and depression were unique influencing factors for the non-smokers: Urban residents had a 1.24 times higher risk of belonging to the simple chronic disease group. As depression levels increased, the risk of severe comorbid chronic disease patterns increased by 1.02 times. Gender and life satisfaction were specific influencing factors for current

**Table 2. Univariate analysis of comorbidity patterns by smoking status.**

| Variable | Group | Smoking Status N(%)/ M±SD | | | | | | | | | | | | | | |
| --- | --- | --- | --- | --- | --- | --- | --- | --- | --- | --- | --- | --- | --- | --- | --- | --- |
| | | Non-smoker (n=6507) | | | | | Current Smoker (n=2873) | | | | | Former Smoker (n=1391) | | | | |
| | | Healthy | Simple | Minor | Major | χ² | Healthy | Simple | Minor | Major | χ² | Healthy | Simple | Minor | Major | χ² |
| Gender | Male | 266 (20.1) | 446 (19.3) | 452 (19.0) | 91 (18.6) | 0.81 | 703 (94.2) | 917 (95.2) | 879 (90.0) | 158 (84.5) | 39.75*** | 174 (95.1) | 347 (91.6) | 591 (91.9) | 161 (86.6) | 9.02* |
| | Female | 1059 (79.9) | 1868 (80.7) | 1926 (81.0) | 399 (81.4) | | 43 (5.8) | 46 (4.8) | 98 (10.0) | 29 (15.5) | | 9 (4.9) | 32 (8.4) | 52 (8.1) | 25 (13.4) | |
| Age (years) | | 60.63±9.09 | 64.14±9.53 | 66.13±9.17 | 68.68±8.49 | 136.88*** | 62.64±8.97 | 64.20±9.07 | 65.93±8.74 | 68.61±8.70 | 32.56*** | 64.90±9.81 | 65.58±8.78 | 68.94±9.06 | 72.01±8.50 | 30.91*** |
| Marital Status | Married | 1165 (87.9) | 1956 (84.5) | 1937 (81.5) | 382 (78.0) | 38.89*** | 661 (88.6) | 853 (88.6) | 842 (86.2) | 151 (80.7) | 10.87* | 165 (90.2) | 341 (90.0) | 562 (87.4) | 157 (84.4) | 4.72 |
| | Other | 160 (12.1) | 358 (15.5) | 441 (18.5) | 108 (22.0) | | 85 (11.4) | 110 (11.4) | 135 (13.8) | 36 (19.3) | | 18 (9.8) | 38 (10.0) | 81 (12.6) | 29 (15.6) | |
| Education Level | Primary School and Below | 830 (62.6) | 1610 (69.6) | 1621 (68.2) | 361 (73.7) | 27.38*** | 385 (51.6) | 528 (54.8) | 591 (60.5) | 121 (64.7) | 19.82*** | 99 (54.1) | 211 (55.7) | 359 (55.8) | 111 (59.7) | 1.31 |
| | Middle School and Above | 495 (37.4) | 704 (30.4) | 757 (31.8) | 129 (26.3) | | 361 (48.4) | 435 (45.2) | 386 (39.5) | 66 (35.3) | | 84 (45.9) | 168 (44.3) | 284 (44.2) | 75 (40.3) | |
| Place of Residence | Rural | 549 (41.4) | 843 (36.4) | 969 (40.7) | 208 (42.4) | 14.53** | 284 (38.1) | 341 (35.4) | 381 (39.0) | 67 (35.8) | 3.03 | 74 (40.4) | 152 (40.1) | 289 (44.9) | 79 (42.5) | 2.76 |
| | Urban | 776 (58.6) | 1471 (63.6) | 1409 (59.3) | 282 (57.6) | | 462 (61.9) | 622 (64.6) | 596 (61.0) | 120 (64.2) | | 109 (59.6) | 227 (59.9) | 354 (55.1) | 107 (57.5) | |
| Alcohol Consumption | No | 981 (74.0) | 1734 (74.9) | 1875 (78.8) | 422 (86.1) | 39.77*** | 250 (33.5) | 370 (38.4) | 432 (44.2) | 101 (54.0) | 36.56*** | 62 (33.9) | 138 (36.4) | 331 (51.5) | 115 (61.8) | 51.20*** |
| | Yes | 344 (26.0) | 580 (25.1) | 503 (21.2) | 68 (13.9) | | 496 (66.5) | 593 (61.6) | 545 (55.8) | 86 (46.0) | | 121 (66.1) | 241 (63.6) | 312 (48.5) | 71 (38.2) | |
| Outpatient | No | 1205 (90.9) | 1869 (80.8) | 1748 (73.5) | 286 (58.4) | 281.42*** | 703 (94.2) | 843 (87.5) | 788 (80.7) | 123 (65.8) | 126.48*** | 175 (95.6) | 307 (81.0) | 464 (72.2) | 104 (55.9) | 88.70*** |
| | Yes | 120 (9.1) | 445 (19.2) | 630 (26.5) | 204 (41.6) | | 43 (5.8) | 120 (12.5) | 189 (19.3) | 64 (34.2) | | 8 (4.4) | 72 (19.0) | 179 (27.8) | 82 (44.1) | |
| Hospitalization | No | 1250 (94.3) | 2022 (87.4) | 1815 (76.3) | 265 (54.1) | 497.88*** | 692 (92.8) | 849 (88.2) | 781 (79.9) | 113 (60.4) | 148.85*** | 170 (92.9) | 317 (83.6) | 438 (68.1) | 82 (44.1) | 142.94*** |
| | Yes | 75 (5.7) | 292 (12.6) | 563 (23.7) | 225 (45.9) | | 54 (7.2) | 114 (11.8) | 196 (20.1) | 74 (39.6) | | 13 (7.1) | 62 (16.4) | 205 (31.9) | 104 (55.9) | |
| Physical Activity Level | Mild | 177 (13.4) | 296 (12.8) | 387 (16.3) | 104 (21.2) | 50.45*** | 104 (13.9) | 155 (16.1) | 136 (13.9) | 38 (20.3) | 24.92*** | 27 (14.8) | 48 (12.7) | 107 (16.6) | 48 (25.8) | 27.71*** |
| | Moderate | 334 (25.2) | 556 (24.0) | 646 (27.2) | 144 (29.4) | | 170 (22.8) | 227 (23.6) | 264 (27.0) | 65 (34.8) | | 44 (24.0) | 116 (30.6) | 214 (33.3) | 63 (33.9) | |
| | Severe | 814 (61.4) | 1462 (63.2) | 1345 (56.6) | 242 (49.4) | | 472 (63.3) | 581 (60.3) | 577 (59.1) | 84 (44.9) | | 112 (61.2) | 215 (56.7) | 322 (50.1) | 75 (40.3) | |
| Sleep Duration(hours) | | 6.44±1.67 | 6.05±1.85 | 5.82±1.92 | 5.29±2.05 | 56.43*** | 6.46±1.63 | 6.30±1.75 | 6.18±1.82 | 5.56±2.01 | 13.60*** | 6.72±1.49 | 6.23±1.71 | 5.94±1.75 | 5.91±2.07 | 10.89*** |
| Cognitive Function | | 13.10±3.18 | 12.85±3.09 | 12.69±3.27 | 12.36±3.32 | 7.98*** | 12.96±3.16 | 12.87±3.13 | 12.86±3.23 | 11.96±3.38 | 5.18** | 13.25±3.15 | 12.76±3.17 | 12.97±3.43 | 12.11±3.58 | 4.22** |
| Depression | | 7.25±5.96 | 8.10±6.16 | 8.14±6.31 | 8.43±6.59 | 7.80*** | 7.46±6.19 | 8.38±6.27 | 7.95±5.95 | 8.89±6.29 | 4.48** | 7.58±6.64 | 8.67±6.36 | 7.85±6.15 | 8.77±6.92 | 2.40 |
| Life Satisfaction | | 3.30±0.70 | 3.28±0.76 | 3.26±0.76 | 3.26±0.79 | 0.76 | 3.35±0.70 | 3.21±0.75 | 3.21±0.72 | 3.25±0.81 | 6.53*** | 3.34±0.75 | 3.27±0.75 | 3.30±0.74 | 3.19±0.71 | 1.44 |

Note: *P<0.05; **P<0.01; ***P<0.001.

**Table 3. Descriptive statistics of disease prevalence by smoking status.**

| Variable | Non-smoker (n=6507) | | | | | Current Smoker (n=2873) | | | | | Former Smoker (n=1391) | | | | |
|---|---|---|---|---|---|---|---|---|---|---|---|---|---|---|---|
| | Simple | Minor | Major | Total | $\chi^2$ | Simple | Minor | Major | Total | $\chi^2$ | Simple | Minor | Major | Total | $\chi^2$ |
| Diabetes | | 704 (29.6) | 250 (51.0) | 954 (18.4) | 1067.45*** | | 242 (24.8) | 92 (49.2) | 334 (15.7) | 398.56*** | | 173 (26.9) | 83 (44.6) | 256 (21.2) | 175.63*** |
| Pulmonary Disease | | 466 (19.6) | 289 (59.0) | 755 (14.6) | 1219.33*** | | 301 (30.8) | 129 (69.0) | 430 (20.2) | 587.71*** | | 179 (27.8) | 119 (64.0) | 298 (24.7) | 282.25*** |
| Heart Disease | | 1018 (42.8) | 393 (80.2) | 1411 (27.2) | 1851.14*** | | 277 (28.4) | 142 (75.9) | 419 (19.7) | 656.35*** | | 210 (32.7) | 143 (76.9) | 353 (29.2) | 364.42*** |
| Stroke | | 224 (9.4) | 165 (33.7) | 389 (7.5) | 683.54*** | | 120 (12.3) | 56 (29.9) | 176 (8.3) | 223.27*** | | 81 (12.6) | 59 (31.7) | 140 (11.6) | 123.89*** |
| Kidney Disease | | 388 (16.3) | 231 (47.1) | 619 (11.9) | 934.24*** | | 192 (19.7) | 95 (50.8) | 287 (13.5) | 404.96*** | | 130 (20.2) | 101 (54.3) | 231 (19.1) | 238.94*** |
| Emotional/Mental Issues | | 80 (3.4) | 83 (16.9) | 163 (3.1) | 381.52*** | | 20 (2.0) | 16 (8.6) | 36 (1.7) | 60.67*** | | 18 (2.8) | 16 (8.6) | 34 (2.8) | 33.75*** |
| Dementia | | 115 (4.8) | 140 (28.6) | 255 (4.9) | 705.60*** | | 44 (4.5) | 46 (24.6) | 90 (4.2) | 234.16*** | | 33 (5.1) | 39 (21.0) | 72 (6.0) | 99.55*** |
| Asthma | | 137 (5.8) | 177 (36.1) | 314 (6.1) | 927.62*** | | 72 (7.4) | 80 (42.8) | 152 (7.1) | 432.04*** | | 50 (7.8) | 81 (43.5) | 131 (10.8) | 258.12*** |
| Hypertension | 974 (42.1) | 1180 (49.6) | 339 (69.2) | 2493 (48.1) | 122.92*** | 406 (42.2) | 433 (44.3) | 127 (67.9) | 966 (45.4) | 42.78*** | 183 (48.3) | 333 (51.8) | 136 (73.1) | 652 (54.0) | 33.62*** |
| Dyslipidemia | 491 (21.2) | 926 (38.9) | 310 (63.3) | 1727 (33.3) | 384.06*** | 195 (20.2) | 284 (29.1) | 92 (49.2) | 571 (26.8) | 71.37*** | 102 (26.9) | 221 (34.4) | 107 (57.5) | 430 (35.6) | 51.91*** |
| Malignant Tumor | 55 (2.4) | 71 (3.0) | 17 (3.5) | 143 (2.8) | 2.64 | 16 (1.7) | 15 (1.5) | 5 (2.7) | 36 (1.7) | 1.08 | 20 (5.3) | 17 (2.6) | 10 (5.4) | 47 (3.9) | 5.72 |
| Liver Disease | 104 (4.5) | 215 (9.0) | 99 (20.2) | 418 (8.1) | 140.21*** | 53 (5.5) | 94 (9.6) | 39 (20.9) | 186 (8.7) | 47.99*** | 23 (6.1) | 71 (11.0) | 32 (17.2) | 126 (10.4) | 17.11*** |
| Stomach/Digestive System | 885 (38.2) | 972 (40.9) | 287 (58.6) | 2144 (41.4) | 69.33*** | 369 (38.3) | 363 (37.2) | 110 (58.8) | 842 (39.6) | 32.00*** | 141 (37.2) | 214 (33.3) | 84 (45.2) | 439 (36.3) | 8.98*** |
| Arthritis/Rheumatism | 1118 (48.3) | 1124 (47.3) | 320 (65.3) | 2562 (49.4) | 55.01*** | 429 (44.5) | 381 (39.0) | 101 (54.0) | 911 (42.8) | 16.57*** | 143 (37.7) | 233 (36.2) | 95 (51.1) | 471 (39.0) | 13.72*** |

Note: * $P<0.05$; ** $P<0.01$; *** $P<0.001$.

**Table 4. Multivariate analysis of comorbidity patterns by smoking status.**

| Variable | Group | Non-smoker (n = 6507) | | | | | |
|---|---|---|---|---|---|---|---|
| | | Simple | | Minor | | Major | |
| | | OR | 95%CI | OR | 95%CI | OR | 95%CI |
| Age(years) | | 1.05*** | 1.04–1.05 | 1.07*** | 1.06–1.07 | 1.09*** | 1.07–1.10 |
| Marital Status | Other | 0.91 | 0.73–1.13 | 0.91 | 0.73–1.12 | 0.81 | 0.60–1.11 |
| Education Level | Middle School and Above | 0.92 | 0.79–1.07 | 1.11 | 0.95–1.30 | 1.04 | 0.80–1.36 |
| Place of Residence | Urban | 1.24** | 1.07–1.44 | 1.09 | 0.94–1.27 | 0.99 | 0.78–1.25 |
| Alcohol Consumption | Yes | 1.06 | 0.90–1.24 | 0.88 | 0.74–1.04 | 0.59** | 0.44–0.80 |
| Hospitalization | Yes | 1.90*** | 1.45–2.48 | 3.59*** | 2.77–4.65 | 7.98*** | 5.87–10.86 |
| Outpatient | Yes | 2.22*** | 1.79–2.76 | 3.03*** | 2.44–3.76 | 4.89*** | 3.71–6.45 |
| Physical Activity Level | Severe | 1.24 | 1.00–1.53 | 0.97 | 0.79–1.20 | 0.80 | 0.58–1.09 |
| | Moderate | 1.12 | 0.89–1.43 | 1.02 | 0.80–1.29 | 0.92 | 0.66–1.30 |
| Sleep Duration(hours) | | 0.90*** | 0.87–0.94 | 0.86*** | 0.82–0.89 | 0.77*** | 0.73–0.82 |
| Cognitive Function | | 0.99 | 0.96–1.01 | 0.97** | 0.95–0.99 | 0.94** | 0.91–0.98 |
| Depression | | 1.02** | 1.01–1.03 | 1.02** | 1.01–1.03 | 1.02* | 1.00–1.04 |
| Current Smoker (n = 2873) | | | | | | | |
| Gender | Female | 0.71 | 0.46–1.11 | 1.48 | 0.99–2.20 | 1.94* | 1.10–3.41 |
| Age(years) | | 1.02** | 1.01–1.03 | 1.04*** | 1.02–1.05 | 1.05*** | 1.03–1.07 |
| Marital Status | Other | 0.87 | 0.64–1.19 | 0.87 | 0.64–1.20 | 1.01 | 0.62–1.64 |
| Education Level | Middle School and Above | 0.94 | 0.77–1.15 | 0.85 | 0.69–1.04 | 0.87 | 0.60–1.26 |
| Alcohol Consumption | Yes | 0.87 | 0.70–1.07 | 0.80* | 0.65–0.99 | 0.67* | 0.46–0.95 |
| Hospitalization | Yes | 1.51* | 1.07–2.13 | 2.59*** | 1.87–3.60 | 5.45*** | 3.56–8.36 |
| Outpatient | Yes | 2.24*** | 1.55–3.23 | 3.38*** | 2.37–4.82 | 6.48*** | 4.11–10.22 |
| Physical Activity Level | Severe | 0.85 | 0.64–1.13 | 1.12 | 0.83–1.51 | 0.73 | 0.45–1.19 |
| | Moderate | 0.91 | 0.66–1.26 | 1.31 | 0.94–1.83 | 1.31 | 0.78–2.17 |
| Sleep Duration(hours) | | 0.95 | 0.89–1.00 | 0.93** | 0.88–0.98 | 0.79*** | 0.72–0.87 |
| Cognitive Function | | 1.00 | 0.97–1.03 | 0.99 | 0.96–1.02 | 0.93** | 0.88–0.98 |
| Depression | | 1.02 | 1.00–1.03 | 1.00 | 0.98–1.02 | 1.02 | 0.99–1.05 |
| Life Satisfaction | | 0.80** | 0.69–0.92 | 0.73*** | 0.64–0.86 | 0.85 | 0.66–1.09 |
| Former Smoker (n = 1391) | | | | | | | |
| Gender | Female | 1.71 | 0.78–3.75 | 1.27 | 0.59–2.75 | 1.99 | 0.83–4.76 |
| Age (years) | | 1.01 | 0.99–1.03 | 1.04*** | 1.02–1.06 | 1.07*** | 1.04–1.10 |
| Alcohol Consumption | Yes | 0.99 | 0.67–1.46 | 0.63* | 0.44–0.92 | 0.57* | 0.35–0.91 |
| Hospitalization | Yes | 2.16* | 1.14–4.09 | 4.17*** | 2.28–7.63 | 9.59*** | 4.96–18.55 |
| Outpatient | Yes | 4.74*** | 2.22–10.12 | 6.72*** | 3.20–14.11 | 11.02*** | 4.99–24.31 |
| Physical Activity Level | Severe | 1.02 | 0.59–1.75 | 0.83 | 0.50–1.39 | 0.53* | 0.29–0.99 |
| | Moderate | 1.42 | 0.78–2.59 | 1.31 | 0.74–2.31 | 0.95 | 0.49–1.86 |
| Sleep Duration(hours) | | 0.85** | 0.76–0.94 | 0.77*** | 0.69–0.86 | 0.80** | 0.71–0.91 |
| Cognitive Function | | 0.95 | 0.90–1.01 | 0.97 | 0.92–1.03 | 0.91** | 0.85–0.97 |

Note: *P < 0.05; **P < 0.01; ***P < 0.001.

smokers: Women had a 1.94 times higher risk of being in the major complex multiple chronic conditions group. An increase in life satisfaction reduced the risk of severe comorbid chronic disease patterns by 20%-27%. Physical activity level was a unique influencing factor for those who had quit smoking: Individuals engaged in intense physical activity had a 47% lower risk of being in the major complex multiple chronic conditions group.

## Discussion

### Current situation analysis

Due to the heterogeneity in research methodologies and definitions of comorbidity, the reported prevalence of multimorbidity in chronic diseases varies significantly across studies. Xu et al.'s meta-analysis reported prevalence rates ranging from 3.5% to 100%, while Violan's study indicated rates between 12.9% and 95.1% [22]. There are also regional differences in prevalence rates, with South America showing the highest prevalence of multimorbidity at 45.7% (95% CI = 39.0–52.5%), followed by North America at 43.1% (95% CI = 32.3–53.8%), Europe at 39.2% (95% CI = 33.2–45.2%), and Asia at 35% (95% CI = 31.4–38.5%) [23]. Middle-aged and elderly individuals, due to factors such as physiological decline, have a higher susceptibility to chronic diseases. Recent domestic studies in China reveal that the prevalence of multimorbidity among middle-aged and elderly patients with chronic diseases is 72.71% [24]. In this study, the prevalence of at least one chronic disease among middle-aged and elderly individuals in China is as high as 79.1%, which closely aligns with the 75% reported in the Healthy China 2019–2030 initiative.

Regarding specific diseases, hypertension, dyslipidemia, stomach/digestive system issues, and arthritis/rheumatism are among the most prevalent chronic diseases in middle-aged and elderly people across all three smoking statuses, a finding nearly identical to that of previous studies [25]. However, a distinct difference in the current smokers group is that lung diseases have surpassed heart disease as the fifth most prevalent condition [26].

### Common influencing factors

Our study found that age is one of the most significant risk factors for the occurrence and progression of chronic diseases. The strong positive correlation between age and the prevalence of multimorbidity has been confirmed in numerous studies. As individuals age, the likelihood of having multiple chronic diseases simultaneously increases significantly [27,28]. Notably, the prevalence of multimorbidity in individuals aged 45 and above increases exponentially compared to those under 45 [29]. This phenomenon often complicates disease management and treatment outcomes.

The relationship between alcohol consumption and the incidence of various chronic diseases has been contentious. Most studies have treated alcohol consumption as a single entity, with the predominant view being that drinking behaviors contribute to the development of chronic diseases [30,31]. In China, a study involving more than 500,000 adults revealed that alcohol consumption is associated with an increased risk of 61 diseases, including liver cirrhosis, stroke, and various types of cancer. The study also highlighted that certain drinking habits, such as daily alcohol consumption, binge drinking, drinking after meals or at irregular times, and consuming spirits, particularly heighten the risk of these diseases [32]. In this study, the risk associated with alcohol consumption in severe multiple chronic condition patterns was reduced, which may align with a scholar's viewpoint suggesting that long-term alcohol consumption in older adults increases the likelihood of fatal diseases like cancer, potentially leading to a masking effect [33]. As scholars delve deeper into this issue, increasing attention is being given to the relationship between the quantity of alcohol consumed, the type of alcoholic beverages, and the development of chronic diseases [34–36]. Research has indicated that beer and spirits are associated with heightened risks for all health outcomes, while only wine has demonstrated a protective effect against ischemic heart disease [37]. Further research is necessary to substantiate these findings.

Sleep problems in middle-aged and elderly populations can be caused by various factors, including psychological, social, demographic, and lifestyle factors, all of which have been

shown to relate to sleep disturbances [38,39]. The findings of this study suggest that increased sleep duration reduces the risk of multiple chronic conditions, aligning with previous research findings [40]. However, many clinical studies specifically examining the relationship between sleep duration and disease have found that this relationship is not simply linear. A systematic review incorporating 225,858 participants aged between 18 and 106 years revealed that both short sleep duration is associated with an increased risk of hypertension (OR: 1.21; 95%CI: 1.09–1.34), and prolonged sleep duration also increases hypertension risk (OR: 1.11; 95%CI: 1.04–1.18) [41]. It has been pointed out that nighttime sleep duration and the risk of cardiovascular or cerebrovascular diseases exhibit a significant U-shaped relationship, with the lowest risk at about 7.5 hours per night [42]. This result is similar to that of Lin et al., where 7.84 hours was identified as the optimal cutoff point for nighttime sleep duration, associated with the lowest probability of developing comorbidities [43]. Nonetheless, it is undeniable that prioritizing personal health habits and optimizing sleep patterns according to individual circumstances are critical for middle-aged and elderly individuals.

Studies have shown that individuals with cognitive impairment tend to have lower health literacy, weaker disease awareness, and significantly reduced self-management capabilities [44]. Poor health self-management may be related to an increase in the number of chronic diseases. In this study, patients with poor cognitive function were more prone to multimorbidity. However, there is limited research supporting the idea that cognitive decline directly leads to an increased probability of chronic disease multimorbidity, which warrants further exploration. Conversely, multiple studies have indicated that the presence of multiple chronic diseases may serve as a risk factor for cognitive impairment in later life. Affected individuals are more likely to experience significant cognitive decline compared to healthy individuals, and the degree of decline is closely related to the number of chronic diseases [45]. This may be due to the cumulative effects of multiple chronic conditions accelerating cognitive deterioration, ultimately leading to cognitive impairment [46].

## Distinct influencing factors

Gender differences in the prevalence of various chronic diseases also exist. For instance, males, possibly due to diet and lifestyle habits, have a higher risk of developing hypertension than females, though the risk levels are not absolute across genders [47]. Studies exploring the risk of diabetes across different age groups show that men aged 35–55 have a higher incidence rate. However, after the age of 55, the incidence rate in women may surpass that of men, possibly due to menopause-related hormonal changes, including decreased estrogen levels, which influence blood sugar levels [48]. In the smoking population of this study, the overall burden of multiple chronic conditions is heavier in women than in men. This aligns with an analysis, which reviewed chronic disease incidence data from 1998 to 2018 and found that women consistently had higher rates of chronic diseases than men, with the gap most pronounced in 2008 when women's chronic disease rates were 45.2 per thousand higher than those of men, although the gap has gradually narrowed over the subsequent decade [49]. This may relate to the physiological characteristics of women and their socio-economic status, highlighting them as a critical demographic for chronic disease prevention and management.

In the middle-aged and elderly population with somatic diseases, depressive symptoms are more prevalent and severe. This may be attributed to structural and functional changes in the brain that predispose older individuals to depression. The findings of this study indicate that depression is a risk factor for multimorbidity in chronic diseases, consistent with other research. A study from Sweden identified the role of depression severity and symptom phenotypes in the progression of somatic comorbidity. In multi-adjusted

models, compared to individuals without depression, those with major depression (β: 0.33, 95% CI: 0.06–0.61) and subsyndromal depression (β: 0.21, 95% CI: 0.12–0.30) experienced a faster accumulation of somatic multimorbidity, while those with minor depression did not [50].

Sedentary behavior (SB) refers to any waking behavior characterized by an energy expenditure of ≤ 1.5 metabolic equivalents (METs) while sitting, reclining, or lying [51]. The majority of chronic diseases are associated with high levels of SB. An Irish scholar conducted a cross-sectional study involving a population sample of 6,903 adults aged 50 and above, revealing a linear relationship between higher levels of sedentary behavior (SB) and the increasing number of chronic disease comorbidities [52]. Promoting physical activity is considered a fundamental strategy for the prevention and management of chronic diseases in middle-aged and elderly individuals [53]. Adequate physical activity offers substantial protection for the physical and mental health of the elderly, significantly reducing the risk of various chronic diseases [54]. Life satisfaction among the elderly serves as a metric for assessing the well-being index of China's older population and is a key component of achieving healthy aging and implementing the Healthy China strategy. In the smoking population, life satisfaction is generally lower, consistent with the findings of this study, and is below that of non-smokers and those who have quit smoking [55]. Life satisfaction, as a subjective evaluation of overall quality of life, reflects an individual's material and spiritual living standards and is linked to complex factors such as health, economic status, and social support [56]. Health, a strong predictor of life satisfaction, has been demonstrated to significantly influence levels of satisfaction.

## Limitations

This study has certain limitations: Firstly, it utilized public databases, restricting the explanatory variables to those available within the data, thus limiting the scope of variables considered. Secondly, variables such as the status of chronic diseases rely on self-reported data, which may introduce recall bias. Additionally, the definition of complex chronic disease grouping is derived from international studies and lacks localization features.

## Conclusion

Based on the latest CHARLS data, this study aims to reveal the distribution of multiple chronic conditions patterns under different smoking statuses and their influencing factors among middle-aged and elderly individuals in China. Approximately 79.1% of this demographic suffers from at least one chronic disease. Multivariate analysis revealed that demographic data, lifestyle, and psychological factors are also significant in influencing the extent of multiple chronic conditions. It is recommended to strengthen public health strategies and interventions related to multiple chronic conditions among smokers to reduce the burden of chronic diseases. Effective policies and interventions are expected to reduce smoking rates across society, thereby lowering the incidence and related mortality of chronic diseases. With the continuous updates and expansion of public databases, it is hoped that future scholars will be able to incorporate more variables to explore the relationships and conduct in-depth analyses of specific comorbidity patterns related to smoking. Cluster analysis could be employed to identify more detailed comorbidity models. Additionally, utilizing multiple waves of the CHARLS study, cohort analysis can be used to examine the long-term trends in chronic diseases among the smoking population in China and to investigate the causal relationships between variables over time.

## Acknowledgments

We would like to express our gratitude to the China Health and Retirement Longitudinal Study (CHARLS) for providing the valuable dataset used in this study.

## Author contributions

**Conceptualization:** Shanwen Sun, Yali Wang.

**Data curation:** Shanwen Sun.

**Formal analysis:** Yali Wang.

**Funding acquisition:** Yali Wang.

**Investigation:** Hailong Hou.

**Methodology:** Hailong Hou, Linlin Chen.

**Project administration:** Linlin Chen, Yuqi Niu.

**Resources:** Yuqi Niu, Yefan Shao.

**Software:** Yuqi Niu, Yefan Shao.

**Supervision:** Xiaochun Zhang.

**Validation:** Xiaochun Zhang.

**Visualization:** Xiaochun Zhang.

**Writing – original draft:** Shanwen Sun.

**Writing – review & editing:** Shanwen Sun, Xiaochun Zhang.

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
