## [Decision Letter · Decision Letter 0]

30 Aug 2024

PONE-D-24-23479Analysis of Chronic Disease Comorbidity Patterns in Middle-aged and Elderly Smokers in China — Based on the China Health and Retirement Longitudinal SurveyPLOS ONE

Dear Dr. Zhang,

Thank you for submitting your manuscript to PLOS ONE. After careful consideration, we feel that it has merit but does not fully meet PLOS ONE’s publication criteria as it currently stands. Therefore, we invite you to submit a revised version of the manuscript that addresses the points raised during the review process.

1. Please revise your manuscript according the comments of the reviewers

We look forward to receiving your revised manuscript.

Kind regards,

Diana Laila Ramatillah, PhD

Academic Editor

PLOS ONE

Additional Editor Comments:

see the comments from the reviewers

Reviewers' comments:

Reviewer's Responses to Questions

**Comments to the Author**

1. Is the manuscript technically sound, and do the data support the conclusions?

Reviewer #1: Yes

Reviewer #2: Partly

Reviewer #3: Yes

Reviewer #4: Yes

Reviewer #5: Yes

2. Has the statistical analysis been performed appropriately and rigorously?

Reviewer #1: Yes

Reviewer #2: No

Reviewer #3: Yes

Reviewer #4: Yes

Reviewer #5: Yes

3. Have the authors made all data underlying the findings in their manuscript fully available?

Reviewer #1: Yes

Reviewer #2: Yes

Reviewer #3: No

Reviewer #4: Yes

Reviewer #5: No

4. Is the manuscript presented in an intelligible fashion and written in standard English?

Reviewer #1: Yes

Reviewer #2: No

Reviewer #3: Yes

Reviewer #4: Yes

Reviewer #5: Yes

5. Review Comments to the Author

Reviewer #1: in the present manuscript entitled "Analysis of Chronic Disease Comorbidity Patterns in Middle-aged and Elderly Smokers

in China — Based on the China Health and Retirement Longitudinal Survey" authors highlighted the very important issues. My submission is Accept it for publications

Reviewer #2: Dear authors,

I have thoroughly reviewed the manuscript titled "Analysis of Chronic Disease Comorbidity Patterns in Middle-aged and Elderly Smokers in China — Based on the China Health and Retirement Longitudinal Survey." The study presents a comprehensive analysis of chronic disease comorbidities among different smoking statuses within the Chinese population. While the manuscript addresses a critical public health issue with significant implications, there are several areas where clarity, writing, and analytical rigor could be enhanced.

The following review provides suggestions to help the author to improve the overall quality and impact of the paper.

Introduction

- Mentioning how smoking specifically contributes to the types of chronic diseases in the introduction would be beneficial.

Methodology

- Include references for the definitions of "Simple," "Mildly Complex," and "Major Complex" chronic disease groups.

- Middle-aged individuals were defined as starting at 45 years. It would be helpful to include a reference supporting this definition.

- The number of informed consents obtained is not clear. The methodology section mentions that two copies of informed consent have been signed; however, in the consent for publication section, additional consent was obtained from some individuals.

Results

- Clarify and verify percentages in tables, ensuring they add up correctly.

- Show statistical measures in the tables, e.g. N (%) or M±SD

Discussion

- It is better to start the discussion with the key findings.

- Expand the discussion by compare your data with other regions and countries studies.

Writing

- Some typos have to be addressed; for example (CHARIS) has been used instead of (CHARLS) in the methods section, and (China’s futu2030”).

- Consider removing the double quotation marks “”.

Reviewer #3: The manuscript presents a well written study of a technically sound scientific enquiry with data that support the research hypothesis. The study was based on a secondary analysis of data extracted from the China Health and Retirement Longitudinal Survey. I made the following observations:

1. The title should be re-written in a sentence case format in accordance with this journal submission guidelines.

2. The authors should provide a more detail Data Availably Statement describing where the datasets can be accessed. They should also clarify if the respondents where de-identified in the data

3. The units of measurements such as age and sleep duration should be clearly defined in all the table of results.

4. Line 93: The authors should include the citations of the previous studies quoted in the statement.

5. Lines 257,261,271,280,300 and 304 contained statements referring to some studies, however only one citation was used in each of them. Addition citations should be provided for these studies, otherwise the statement should be rewritten.

6. The authors should further explain the nature of the missing variables and whether the deletion approach was the appropriate method to handle the missing values in the data. They should provide the total number respondents that met the inclusion criteria and percentage of the missing values calculated.

7. The authors should include a supporting information figure flow charts of participants.

8. They should provide directions for future research in this area based on their findings.

Reviewer #4: The article offers a comprehensive review of Chronic Disease Comorbidity Patterns and Smoking in China. The paper is well-structured and detailed. However, there are some areas in the Abstract that require further clarification and detail.

Firstly, the Abstract is missing essential information in the Method and Result sections. The Method section does not include the sample size, nor does it explain how the samples were acquired. At least it would be beneficial to mention that the data was derived from the CHARIS survey, as this adds context to the methodology.

Secondly, the way the results are presented in the Abstract lacks clarity and does not effectively convey the findings. I found it challenging to grasp the key outcomes without reading the entire paper. Below are specific sentences from the Results section that raised questions upon my initial reading of the Abstract:

• "Age, alcohol consumption, hospital admissions in the past year, outpatient visits in the past month, sleep, and cognitive function were common influencing factors." Influencing factors for what?

• "Place of residence and depression were unique influencing factors for the non-smoking group." Again, influencing what?

• "Gender and life satisfaction were specific to the smoking group." Specific to what?

• "The level of physical activity was a unique factor for the group that had quit smoking." Unique for what?

Furthermore, the paper does not consistently mention Relative Risk (RR) or Confidence Intervals (CI) when comparing figures. CI, in particular, is omitted throughout the paper, which diminishes the robustness of the statistical analysis. For example:

• "Individuals who consumed alcohol had a 20%-43% lower risk of severe comorbid chronic disease patterns."

• "Patients who had been hospitalized in the past year had a 1.51-9.59 times higher risk of severe comorbid chronic disease patterns."

• "Improvements in cognitive function reduced the risk of severe comorbid chronic disease patterns by 3%-9%."

• "Urban residents had a 1.24 times higher risk of belonging to the simple chronic disease group."

• "As depression levels increased, the risk of severe comorbid chronic disease patterns increased by 1.02 times."

In the Discussion section, the following statements require citations:

• "The relationship between tobacco and respiratory diseases has been extensively validated through research. Tobacco smoke contains a plethora of harmful chemicals, such as polycyclic aromatic hydrocarbons, free radicals, carbon monoxide, and nicotine, which directly damage alveolar and bronchial epithelial cells, leading to structural and functional impairments. Increased smoking leads to decreased efficiency in pulmonary gas exchange, reduced ciliary motility, and heightened risk of infection, thus being a primary cause of respiratory diseases."

Additionally, in the third paragraph of the Discussion, which addresses secondhand smoke, I recommend removing this section. The paper does not focus on secondhand smoke, nor does it address the issue in detail. However, if you decide to retain it, I will respect that decision.

Reviewer #5: For the methodology part, explain in detail about the inclusion and exclusion criteria for your participants

For the discussion part, focus on comparing what your study has found versus what the existing literature has been shown. No need for over-explanation.

6. PLOS authors have the option to publish the peer review history of their article (what does this mean? ). If published, this will include your full peer review and any attached files.

**Do you want your identity to be public for this peer review?** For information about this choice, including consent withdrawal, please see our Privacy Policy .

Reviewer #1: **Yes: ** Dr. Muhammad Kashif

Reviewer #2: No

Reviewer #3: **Yes: ** Abbas Lawal Ibrahim

Reviewer #4: **Yes: ** Alexander M. Ibrahim

Reviewer #5: No

---

## [Author Response · Author response to Decision Letter 1]

20 Sep 2024

Dear Experts：

We sincerely thank the editor and all reviewers for valuable feedback that we

have used to improve the quality of our manuscript.

After careful revision, we have resubmitted the manuscript with specific modifications highlighted for clearer review by the experts. Given the diverse feedback provided by Reviewers 2-4, we have highlighted their comments in different colors for clarity. We have also addressed the feedback from Reviewers 1 and 5 with detailed responses and corresponding line number annotations. The revised paragraphs based on REVIEWER #2 are labeled in yellow. The revised paragraphs based on REVIEWER #3 are labeled in blue. The revised paragraphs based on REVIEWER #4 are labeled in green.

It is our sincere intention that this revised version aligns with your expectations. We highly value this submission and kindly request that you carefully consider our

manuscript. We eagerly anticipate further discussion with you. The specific

modifications are listed as follows:

Reviewer #1: in the present manuscript entitled "Analysis of Chronic Disease Comorbidity Patterns in Middle-aged and Elderly Smokers

in China — Based on the China Health and Retirement Longitudinal Survey" authors highlighted the very important issues. My submission is Accept it for publications.

Thank you for your positive feedback. Based on the suggestions from other reviewers, we have made further revisions to the manuscript. We will continue to approach our research with a responsible, meticulous, and scientifically rigorous attitude. Once again, we appreciate your encouragement and recognition. Please feel free to reach out if you have any further comments or suggestions.

Reviewer #2:

1. Introduction

- Mentioning how smoking specifically contributes to the types of chronic diseases in the introduction would be beneficial.

A: Thank you for your feedback. I apologize for not providing a clear explanation of this section. I have now elaborated on the harmful effects of smoking on various chronic diseases in the background section. Line 89-101

2. Methodology

- Include references for the definitions of "Simple," "Mildly Complex," and "Major Complex" chronic disease groups.

- Middle-aged individuals were defined as starting at 45 years. It would be helpful to include a reference supporting this definition.

A: Thank you for your meticulous observation. I have supplemented the references accordingly. The selection of 45 years as the lower age limit is primarily based on the classification of middle-aged and elderly populations in other studies. Additionally, the target group of the CHARLS data is individuals aged 45 and above, which further supports the inclusion of this age range in our analysis. Line 118

- The number of informed consents obtained is not clear. The methodology section mentions that two copies of informed consent have been signed; however, in the consent for publication section, additional consent was obtained from some individuals.

A: Thank you for your suggestion. We revisited the CHARLS database website and confirmed that the official survey team collected two informed consent forms, as mentioned in the methodology section. In line with the PLOS ONE formatting requirements, we have removed the unclear content from the 'Consent for Publication' section. Line 126

3. Results

- Clarify and verify percentages in tables, ensuring they add up correctly.

- Show statistical measures in the tables, e.g. N (%) or M±SD

A: Thank you for your reminder. We have re-verified all the data and found no errors. Additionally, we have made improvements based on your suggestions. Table1-4

4. Discussion

- It is better to start the discussion with the key findings.

- Expand the discussion by compare your data with other regions and countries studies.

A: Thank you for this valuable suggestion. I apologize for any confusion caused by the section we discussed. Based on the feedback from other reviewers, we have reorganized, trimmed, and supplemented the discussion section, prioritizing the key findings and comparing them with other studies. Line 237

5.Writing

- Some typos have to be addressed; for example (CHARIS) has been used instead of (CHARLS) in the methods section, and (China’s futu2030”).

- Consider removing the double quotation marks “”.

A: We sincerely apologize for our oversight. The necessary corrections have been made, and we will approach our future research with greater caution and attention to detail. Line 67; Line 125

Reviewer #3:

1. The title should be re-written in a sentence case format in accordance with this journal submission guidelines.

A: We apologize for overlooking the formatting requirements. The necessary corrections have now been made. Line 1

2. The authors should provide a more detail Data Availably Statement describing where the datasets can be accessed. They should also clarify if the respondents where de-identified in the data

A: Thank you for your feedback. We have referred to expressions used in other high-impact publications and revised the methodology section to ensure clarity in the description of data collection. Line 125

3.The units of measurements such as age and sleep duration should be clearly defined in all the table of results.

A: Thank you for your feedback. We have added the relevant units to the general data section of the methodology. Table1-4

4. Line 93: The authors should include the citations of the previous studies quoted in the statement.

A: Thank you for your careful observation. We have supplemented this section with the appropriate references. Line 106

5. Lines 257,261,271,280,300 and 304 contained statements referring to some studies, however only one citation was used in each of them. Addition citations should be provided for these studies, otherwise the statement should be rewritten.

A: Based on the feedback from other reviewers, we have revised the discussion section and supplemented it with additional references. Line 263

6. The authors should further explain the nature of the missing variables and whether the deletion approach was the appropriate method to handle the missing values in the data. They should provide the total number respondents that met the inclusion criteria and percentage of the missing values calculated.

A: Given the large sample size of the original database, which is sufficient to support subsequent analyses, we opted for listwise deletion to handle missing data instead of imputation to ensure data authenticity. This approach is also consistent with methods employed in other high-impact studies, as cited in the manuscript. Additionally, since our target population includes all individuals aged 45 and above, no additional inclusion or exclusion criteria were applied. We followed similar practices from other secondary analyses of databases, as referenced. Line 168

7. The authors should include a supporting information figure flow charts of participants.

A: We have visualized the selection process of the study participants. According to the PLOS ONE guidelines, the images have been submitted through the online system.

8. They should provide directions for future research in this area based on their findings.

A: Thank you for your suggestion. This was an oversight on our part. We have now included a discussion on future directions in the conclusion section, expressing our hope that subsequent research can build on this foundation and further enrich and improve the findings. Line 361

Reviewer #4:

1. Firstly, the Abstract is missing essential information in the Method and Result sections. The Method section does not include the sample size, nor does it explain how the samples were acquired. At least it would be beneficial to mention that the data was derived from the CHARIS survey, as this adds context to the methodology.

A: Thank you for your suggestion. We acknowledge that the writing of the abstract was not sufficiently detailed. We have made the necessary additions and adjustments to ensure that the abstract is complete and accurate, allowing readers to quickly and clearly grasp the content of the full article. Line 43

2.Secondly, the way the results are presented in the Abstract lacks clarity and does not effectively convey the findings. I found it challenging to grasp the key outcomes without reading the entire paper. Below are specific sentences from the Results section that raised questions upon my initial reading of the Abstract:

• "Age, alcohol consumption, hospital admissions in the past year, outpatient visits in the past month, sleep, and cognitive function were common influencing factors." Influencing factors for what?

• "Place of residence and depression were unique influencing factors for the non-smoking group." Again, influencing what?

• "Gender and life satisfaction were specific to the smoking group." Specific to what?

• "The level of physical activity was a unique factor for the group that had quit smoking." Unique for what?

A: We sincerely apologize for the confusion caused by this section. This issue may have arisen due to our limited proficiency in English, leading to expressions that may have been influenced by our native language and are difficult to understand in an English-speaking context. We have enlisted the help of a graduate student specializing in translation studies to assist with language polishing. If there are still passages that you find unclear, please feel free to point them out, and we will seek assistance from a professional editing service. Line 48

3.Furthermore, the paper does not consistently mention Relative Risk (RR) or Confidence Intervals (CI) when comparing figures. CI, in particular, is omitted throughout the paper, which diminishes the robustness of the statistical analysis. For example:

• "Individuals who consumed alcohol had a 20%-43% lower risk of severe comorbid chronic disease patterns."

• "Patients who had been hospitalized in the past year had a 1.51-9.59 times higher risk of severe comorbid chronic disease patterns."

• "Improvements in cognitive function reduced the risk of severe comorbid chronic disease patterns by 3%-9%."

• "Urban residents had a 1.24 times higher risk of belonging to the simple chronic disease group."

• "As depression levels increased, the risk of severe comorbid chronic disease patterns increased by 1.02 times."

A: Thank you for your careful observation. We have supplemented all confidence intervals in Table 4 to enhance the statistical power of the analysis. Table 4

4.In the Discussion section, the following statements require citations:

• "The relationship between tobacco and respiratory diseases has been extensively validated through research. Tobacco smoke contains a plethora of harmful chemicals, such as polycyclic aromatic hydrocarbons, free radicals, carbon monoxide, and nicotine, which directly damage alveolar and bronchial epithelial cells, leading to structural and functional impairments. Increased smoking leads to decreased efficiency in pulmonary gas exchange, reduced ciliary motility, and heightened risk of infection, thus being a primary cause of respiratory diseases."

A: After considering the feedback from other reviewers, we realized that we had over-explained the detailed mechanisms and principles between variables. However, the reviewers emphasized focusing more on comparisons with other studies. Therefore, we have removed this section and hope for your understanding.

5.Additionally, in the third paragraph of the Discussion, which addresses secondhand smoke, I recommend removing this section. The paper does not focus on secondhand smoke, nor does it address the issue in detail. However, if you decide to retain it, I will respect that decision.

A: Thank you for your feedback. Our original intention was to provide a more comprehensive discussion of the harmful effects of smoking. However, after careful consideration of your comments, we have removed that section. We have chosen to focus more on the comorbidity of smoking and chronic diseases to emphasize the core content.

Reviewer #5:

1.For the methodology part, explain in detail about the inclusion and exclusion criteria for your participants

A: Thank you for your feedback. We have referred to numerous secondary analyses of public databases, particularly those using the CHARLS database. Such studies typically do not have fixed inclusion or exclusion criteria, as the research theme targets the entire elderly population in China. The key research variables are included in the analysis as long as there are no missing values.

2.For the discussion part, focus on comparing what your study has found versus what the existing literature has been shown. No need for over-explanation.

A: Thank you for your suggestion. We have also recognized this point and, after considering the feedback from other reviewers, revised, trimmed, and improved the discussion section. We have reduced excessive explanations and shifted the focus towards comparing our findings with other studies. The revised discussion emphasizes the key exploration of comorbidity patterns under different smoking statuses and their related factors. Line 237

These are my responses to all the reviewers' comments. I would like to express my sincere gratitude to the experts for their valuable input. Moving forward, I will approach my research with even greater diligence and enthusiasm. If there are any further comments or suggestions, I warmly welcome continued discussion and exchange.

---

## [Editor Report · Decision Letter 1]

8 Oct 2024

PONE-D-24-23479R1Analysis of chronic disease comorbidity patterns in middle-aged and elderly smokers in China: The China Health and Retirement Longitudinal StudyPLOS ONE

Dear Dr. Zhang,

Thank you for submitting your manuscript to PLOS ONE. After careful consideration, we feel that it has merit but does not fully meet PLOS ONE’s publication criteria as it currently stands. Therefore, we invite you to submit a revised version of the manuscript that addresses the points raised during the review process.

We look forward to receiving your revised manuscript.

Kind regards,

Diana Laila Ramatillah, PhD

Academic Editor

PLOS ONE

**Journal Requirements:**

**Additional Editor Comments:**

Please revise the manuscript based on the comments

---

## [Author Response · Author response to Decision Letter 2]

12 Oct 2024

Dear Experts：

We sincerely thank the editor and all reviewers for valuable feedback that we

have used to improve the quality of our manuscript. The reviewer comments are laid

out below in italicized font. Our response is given in normal font.

After careful revision, we have resubmitted the manuscript with specific modifications highlighted for clearer review by the experts. Given the diverse feedback provided by Reviewers 2-4, we have highlighted their comments in different colors for clarity. We have also addressed the feedback from Reviewers 1 and 5 with detailed responses and corresponding line number annotations. The revised paragraphs based on REVIEWER #2 are labeled in yellow. The revised paragraphs based on REVIEWER #3 are labeled in blue. The revised paragraphs based on REVIEWER #4 are labeled in green.

It is our sincere intention that this revised version aligns with your expectations. We highly value this submission and kindly request that you carefully consider our

manuscript. We eagerly anticipate further discussion with you. The specific

modifications are listed as follows:

Reviewer #1: in the present manuscript entitled "Analysis of Chronic Disease Comorbidity Patterns in Middle-aged and Elderly Smokers

in China — Based on the China Health and Retirement Longitudinal Survey" authors highlighted the very important issues. My submission is Accept it for publications.

Thank you for your positive feedback. Based on the suggestions from other reviewers, we have made further revisions to the manuscript. We will continue to approach our research with a responsible, meticulous, and scientifically rigorous attitude. Once again, we appreciate your encouragement and recognition. Please feel free to reach out if you have any further comments or suggestions.

Reviewer #2:

1. Introduction

- Mentioning how smoking specifically contributes to the types of chronic diseases in the introduction would be beneficial.

A: Thank you for your feedback. I apologize for not providing a clear explanation of this section. I have now elaborated on the harmful effects of smoking on various chronic diseases in the background section. Line 89-101

2. Methodology

- Include references for the definitions of "Simple," "Mildly Complex," and "Major Complex" chronic disease groups.

- Middle-aged individuals were defined as starting at 45 years. It would be helpful to include a reference supporting this definition.

A: Thank you for your meticulous observation. I have supplemented the references accordingly. The selection of 45 years as the lower age limit is primarily based on the classification of middle-aged and elderly populations in other studies. Additionally, the target group of the CHARLS data is individuals aged 45 and above, which further supports the inclusion of this age range in our analysis. Line 118

- The number of informed consents obtained is not clear. The methodology section mentions that two copies of informed consent have been signed; however, in the consent for publication section, additional consent was obtained from some individuals.

A: Thank you for your suggestion. We revisited the CHARLS database website and confirmed that the official survey team collected two informed consent forms, as mentioned in the methodology section. In line with the PLOS ONE formatting requirements, we have removed the unclear content from the 'Consent for Publication' section. Line 126

3. Results

- Clarify and verify percentages in tables, ensuring they add up correctly.

- Show statistical measures in the tables, e.g. N (%) or M±SD

A: Thank you for your reminder. We have re-verified all the data and found no errors. Additionally, we have made improvements based on your suggestions. Table1-4

4. Discussion

- It is better to start the discussion with the key findings.

- Expand the discussion by compare your data with other regions and countries studies.

A: Thank you for this valuable suggestion. I apologize for any confusion caused by the section we discussed. Based on the feedback from other reviewers, we have reorganized, trimmed, and supplemented the discussion section, prioritizing the key findings and comparing them with other studies. Line 237

5.Writing

- Some typos have to be addressed; for example (CHARIS) has been used instead of (CHARLS) in the methods section, and (China’s futu2030”).

- Consider removing the double quotation marks “”.

A: We sincerely apologize for our oversight. The necessary corrections have been made, and we will approach our future research with greater caution and attention to detail. Line 67; Line 125

Reviewer #3:

1. The title should be re-written in a sentence case format in accordance with this journal submission guidelines.

A: We apologize for overlooking the formatting requirements. The necessary corrections have now been made. Line 1

2. The authors should provide a more detail Data Availably Statement describing where the datasets can be accessed. They should also clarify if the respondents where de-identified in the data

A: Thank you for your feedback. We have referred to expressions used in other high-impact publications and revised the methodology section to ensure clarity in the description of data collection. Line 125

3.The units of measurements such as age and sleep duration should be clearly defined in all the table of results.

A: Thank you for your feedback. We have added the relevant units to the general data section of the methodology. Table1-4

4. Line 93: The authors should include the citations of the previous studies quoted in the statement.

A: Thank you for your careful observation. We have supplemented this section with the appropriate references. Line 106

5. Lines 257,261,271,280,300 and 304 contained statements referring to some studies, however only one citation was used in each of them. Addition citations should be provided for these studies, otherwise the statement should be rewritten.

A: Based on the feedback from other reviewers, we have revised the discussion section and supplemented it with additional references. Line 263

6. The authors should further explain the nature of the missing variables and whether the deletion approach was the appropriate method to handle the missing values in the data. They should provide the total number respondents that met the inclusion criteria and percentage of the missing values calculated.

A: Given the large sample size of the original database, which is sufficient to support subsequent analyses, we opted for listwise deletion to handle missing data instead of imputation to ensure data authenticity. This approach is also consistent with methods employed in other high-impact studies, as cited in the manuscript. Additionally, since our target population includes all individuals aged 45 and above, no additional inclusion or exclusion criteria were applied. We followed similar practices from other secondary analyses of databases, as referenced. Line 168

7. The authors should include a supporting information figure flow charts of participants.

A: We have visualized the selection process of the study participants. According to the PLOS ONE guidelines, the images have been submitted through the online system.

8. They should provide directions for future research in this area based on their findings.

A: Thank you for your suggestion. This was an oversight on our part. We have now included a discussion on future directions in the conclusion section, expressing our hope that subsequent research can build on this foundation and further enrich and improve the findings. Line 361

Reviewer #4:

1. Firstly, the Abstract is missing essential information in the Method and Result sections. The Method section does not include the sample size, nor does it explain how the samples were acquired. At least it would be beneficial to mention that the data was derived from the CHARIS survey, as this adds context to the methodology.

A: Thank you for your suggestion. We acknowledge that the writing of the abstract was not sufficiently detailed. We have made the necessary additions and adjustments to ensure that the abstract is complete and accurate, allowing readers to quickly and clearly grasp the content of the full article. Line 43

2.Secondly, the way the results are presented in the Abstract lacks clarity and does not effectively convey the findings. I found it challenging to grasp the key outcomes without reading the entire paper. Below are specific sentences from the Results section that raised questions upon my initial reading of the Abstract:

• "Age, alcohol consumption, hospital admissions in the past year, outpatient visits in the past month, sleep, and cognitive function were common influencing factors." Influencing factors for what?

• "Place of residence and depression were unique influencing factors for the non-smoking group." Again, influencing what?

• "Gender and life satisfaction were specific to the smoking group." Specific to what?

• "The level of physical activity was a unique factor for the group that had quit smoking." Unique for what?

A: We sincerely apologize for the confusion caused by this section. This issue may have arisen due to our limited proficiency in English, leading to expressions that may have been influenced by our native language and are difficult to understand in an English-speaking context. We have enlisted the help of a graduate student specializing in translation studies to assist with language polishing. If there are still passages that you find unclear, please feel free to point them out, and we will seek assistance from a professional editing service. Line 48

3.Furthermore, the paper does not consistently mention Relative Risk (RR) or Confidence Intervals (CI) when comparing figures. CI, in particular, is omitted throughout the paper, which diminishes the robustness of the statistical analysis. For example:

• "Individuals who consumed alcohol had a 20%-43% lower risk of severe comorbid chronic disease patterns."

• "Patients who had been hospitalized in the past year had a 1.51-9.59 times higher risk of severe comorbid chronic disease patterns."

• "Improvements in cognitive function reduced the risk of severe comorbid chronic disease patterns by 3%-9%."

• "Urban residents had a 1.24 times higher risk of belonging to the simple chronic disease group."

• "As depression levels increased, the risk of severe comorbid chronic disease patterns increased by 1.02 times."

A: Thank you for your careful observation. We have supplemented all confidence intervals in Table 4 to enhance the statistical power of the analysis. Table 4

4.In the Discussion section, the following statements require citations:

• "The relationship between tobacco and respiratory diseases has been extensively validated through research. Tobacco smoke contains a plethora of harmful chemicals, such as polycyclic aromatic hydrocarbons, free radicals, carbon monoxide, and nicotine, which directly damage alveolar and bronchial epithelial cells, leading to structural and functional impairments. Increased smoking leads to decreased efficiency in pulmonary gas exchange, reduced ciliary motility, and heightened risk of infection, thus being a primary cause of respiratory diseases."

A: After considering the feedback from other reviewers, we realized that we had over-explained the detailed mechanisms and principles between variables. However, the reviewers emphasized focusing more on comparisons with other studies. Therefore, we have removed this section and hope for your understanding.

5.Additionally, in the third paragraph of the Discussion, which addresses secondhand smoke, I recommend removing this section. The paper does not focus on secondhand smoke, nor does it address the issue in detail. However, if you decide to retain it, I will respect that decision.

A: Thank you for your feedback. Our original intention was to provide a more comprehensive discussion of the harmful effects of smoking. However, after careful consideration of your comments, we have removed that section. We have chosen to focus more on the comorbidity of smoking and chronic diseases to emphasize the core content.

Reviewer #5:

1.For the methodology part, explain in detail about the inclusion and exclusion criteria for your participants

A: Thank you for your feedback. We have referred to numerous secondary analyses of public databases, particularly those using the CHARLS database. Such studies typically do not have fixed inclusion or exclusion criteria, as the research theme targets the entire elderly population in China. The key research variables are included in the analysis as long as there are no missing values.

2.For the discussion part, focus on comparing what your study has found versus what the existing literature has been shown. No need for over-explanation.

A: Thank you for your suggestion. We have also recognized this point and, after considering the feedback from other reviewers, revised, trimmed, and improved the discussion section. We have reduced excessive explanations and shifted the focus towards comparing our findings with other studies. The revised discussion emphasizes the key exploration of comorbidity patterns under different smoking statuses and their related factors. Line 237

These are my responses to all the reviewers' comments. I would like to express my sincere gratitude to the experts for their valuable input. Moving forward, I will approach my research with even greater diligence and enthusiasm. If there are any further comments or suggestions, I warmly welcome continued discussion and exchange.

---

## [Editor Report · Decision Letter 2]

30 Oct 2024

PONE-D-24-23479R2Analysis of chronic disease comorbidity patterns in middle-aged and elderly smokers in China: The China Health and Retirement Longitudinal StudyPLOS ONE

Dear Dr. Zhang,

Thank you for submitting your manuscript to PLOS ONE. After careful consideration, we feel that it has merit but does not fully meet PLOS ONE’s publication criteria as it currently stands. Therefore, we invite you to submit a revised version of the manuscript that addresses the points raised during the review process.

We look forward to receiving your revised manuscript.

Kind regards,

Diana Laila Ramatillah, PhD

Academic Editor

PLOS ONE

Journal Requirements:

Additional Editor Comments:

We can accept this manuscript after minor revision

---

## [Author Response · Author response to Decision Letter 3]

20 Nov 2024

Dear Experts：

We sincerely thank the editor and all reviewers for valuable feedback that we have used to improve the quality of our manuscript. However, we have checked that there are no references that have been withdrawn. Meanwhile, we have modified the picture through the inspection of the pace system, and now the format requirements of the picture have met the standard. If there is any problem, please tell me again.

After careful revision, we have resubmitted the manuscript with specific modifications highlighted for clearer review by the experts. Given the diverse feedback provided by Reviewers 2-4, we have highlighted their comments in different colors for clarity. We have also addressed the feedback from Reviewers 1 and 5 with detailed responses and corresponding line number annotations. The revised paragraphs based on REVIEWER #2 are labeled in yellow. The revised paragraphs based on REVIEWER #3 are labeled in blue. The revised paragraphs based on REVIEWER #4 are labeled in green.

It is our sincere intention that this revised version aligns with your expectations. We highly value this submission and kindly request that you carefully consider our

manuscript. We eagerly anticipate further discussion with you. The specific

modifications are listed as follows:

Reviewer #1: in the present manuscript entitled "Analysis of Chronic Disease Comorbidity Patterns in Middle-aged and Elderly Smokers

in China — Based on the China Health and Retirement Longitudinal Survey" authors highlighted the very important issues. My submission is Accept it for publications.

Thank you for your positive feedback. Based on the suggestions from other reviewers, we have made further revisions to the manuscript. We will continue to approach our research with a responsible, meticulous, and scientifically rigorous attitude. Once again, we appreciate your encouragement and recognition. Please feel free to reach out if you have any further comments or suggestions.

Reviewer #2:

1. Introduction

- Mentioning how smoking specifically contributes to the types of chronic diseases in the introduction would be beneficial.

A: Thank you for your feedback. I apologize for not providing a clear explanation of this section. I have now elaborated on the harmful effects of smoking on various chronic diseases in the background section. Line 89-101

2. Methodology

- Include references for the definitions of "Simple," "Mildly Complex," and "Major Complex" chronic disease groups.

- Middle-aged individuals were defined as starting at 45 years. It would be helpful to include a reference supporting this definition.

A: Thank you for your meticulous observation. I have supplemented the references accordingly. The selection of 45 years as the lower age limit is primarily based on the classification of middle-aged and elderly populations in other studies. Additionally, the target group of the CHARLS data is individuals aged 45 and above, which further supports the inclusion of this age range in our analysis. Line 118

- The number of informed consents obtained is not clear. The methodology section mentions that two copies of informed consent have been signed; however, in the consent for publication section, additional consent was obtained from some individuals.

A: Thank you for your suggestion. We revisited the CHARLS database website and confirmed that the official survey team collected two informed consent forms, as mentioned in the methodology section. In line with the PLOS ONE formatting requirements, we have removed the unclear content from the 'Consent for Publication' section. Line 126

3. Results

- Clarify and verify percentages in tables, ensuring they add up correctly.

- Show statistical measures in the tables, e.g. N (%) or M±SD

A: Thank you for your reminder. We have re-verified all the data and found no errors. Additionally, we have made improvements based on your suggestions. Table1-4

4. Discussion

- It is better to start the discussion with the key findings.

- Expand the discussion by compare your data with other regions and countries studies.

A: Thank you for this valuable suggestion. I apologize for any confusion caused by the section we discussed. Based on the feedback from other reviewers, we have reorganized, trimmed, and supplemented the discussion section, prioritizing the key findings and comparing them with other studies. Line 237

5.Writing

- Some typos have to be addressed; for example (CHARIS) has been used instead of (CHARLS) in the methods section, and (China’s futu2030”).

- Consider removing the double quotation marks “”.

A: We sincerely apologize for our oversight. The necessary corrections have been made, and we will approach our future research with greater caution and attention to detail. Line 67; Line 125

Reviewer #3:

1. The title should be re-written in a sentence case format in accordance with this journal submission guidelines.

A: We apologize for overlooking the formatting requirements. The necessary corrections have now been made. Line 1

2. The authors should provide a more detail Data Availably Statement describing where the datasets can be accessed. They should also clarify if the respondents where de-identified in the data

A: Thank you for your feedback. We have referred to expressions used in other high-impact publications and revised the methodology section to ensure clarity in the description of data collection. Line 125

3.The units of measurements such as age and sleep duration should be clearly defined in all the table of results.

A: Thank you for your feedback. We have added the relevant units to the general data section of the methodology. Table1-4

4. Line 93: The authors should include the citations of the previous studies quoted in the statement.

A: Thank you for your careful observation. We have supplemented this section with the appropriate references. Line 106

5. Lines 257,261,271,280,300 and 304 contained statements referring to some studies, however only one citation was used in each of them. Addition citations should be provided for these studies, otherwise the statement should be rewritten.

A: Based on the feedback from other reviewers, we have revised the discussion section and supplemented it with additional references. Line 263

6. The authors should further explain the nature of the missing variables and whether the deletion approach was the appropriate method to handle the missing values in the data. They should provide the total number respondents that met the inclusion criteria and percentage of the missing values calculated.

A: Given the large sample size of the original database, which is sufficient to support subsequent analyses, we opted for listwise deletion to handle missing data instead of imputation to ensure data authenticity. This approach is also consistent with methods employed in other high-impact studies, as cited in the manuscript. Additionally, since our target population includes all individuals aged 45 and above, no additional inclusion or exclusion criteria were applied. We followed similar practices from other secondary analyses of databases, as referenced. Line 168

7. The authors should include a supporting information figure flow charts of participants.

A: We have visualized the selection process of the study participants. According to the PLOS ONE guidelines, the images have been submitted through the online system.

8. They should provide directions for future research in this area based on their findings.

A: Thank you for your suggestion. This was an oversight on our part. We have now included a discussion on future directions in the conclusion section, expressing our hope that subsequent research can build on this foundation and further enrich and improve the findings. Line 361

Reviewer #4:

1. Firstly, the Abstract is missing essential information in the Method and Result sections. The Method section does not include the sample size, nor does it explain how the samples were acquired. At least it would be beneficial to mention that the data was derived from the CHARIS survey, as this adds context to the methodology.

A: Thank you for your suggestion. We acknowledge that the writing of the abstract was not sufficiently detailed. We have made the necessary additions and adjustments to ensure that the abstract is complete and accurate, allowing readers to quickly and clearly grasp the content of the full article. Line 43

2.Secondly, the way the results are presented in the Abstract lacks clarity and does not effectively convey the findings. I found it challenging to grasp the key outcomes without reading the entire paper. Below are specific sentences from the Results section that raised questions upon my initial reading of the Abstract:

• "Age, alcohol consumption, hospital admissions in the past year, outpatient visits in the past month, sleep, and cognitive function were common influencing factors." Influencing factors for what?

• "Place of residence and depression were unique influencing factors for the non-smoking group." Again, influencing what?

• "Gender and life satisfaction were specific to the smoking group." Specific to what?

• "The level of physical activity was a unique factor for the group that had quit smoking." Unique for what?

A: We sincerely apologize for the confusion caused by this section. This issue may have arisen due to our limited proficiency in English, leading to expressions that may have been influenced by our native language and are difficult to understand in an English-speaking context. We have enlisted the help of a graduate student specializing in translation studies to assist with language polishing. If there are still passages that you find unclear, please feel free to point them out, and we will seek assistance from a professional editing service. Line 48

3.Furthermore, the paper does not consistently mention Relative Risk (RR) or Confidence Intervals (CI) when comparing figures. CI, in particular, is omitted throughout the paper, which diminishes the robustness of the statistical analysis. For example:

• "Individuals who consumed alcohol had a 20%-43% lower risk of severe comorbid chronic disease patterns."

• "Patients who had been hospitalized in the past year had a 1.51-9.59 times higher risk of severe comorbid chronic disease patterns."

• "Improvements in cognitive function reduced the risk of severe comorbid chronic disease patterns by 3%-9%."

• "Urban residents had a 1.24 times higher risk of belonging to the simple chronic disease group."

• "As depression levels increased, the risk of severe comorbid chronic disease patterns increased by 1.02 times."

A: Thank you for your careful observation. We have supplemented all confidence intervals in Table 4 to enhance the statistical power of the analysis. Table 4

4.In the Discussion section, the following statements require citations:

• "The relationship between tobacco and respiratory diseases has been extensively validated through research. Tobacco smoke contains a plethora of harmful chemicals, such as polycyclic aromatic hydrocarbons, free radicals, carbon monoxide, and nicotine, which directly damage alveolar and bronchial epithelial cells, leading to structural and functional impairments. Increased smoking leads to decreased efficiency in pulmonary gas exchange, reduced ciliary motility, and heightened risk of infection, thus being a primary cause of respiratory diseases."

A: After considering the feedback from other reviewers, we realized that we had over-explained the detailed mechanisms and principles between variables. However, the reviewers emphasized focusing more on comparisons with other studies. Therefore, we have removed this section and hope for your understanding.

5.Additionally, in the third paragraph of the Discussion, which addresses secondhand smoke, I recommend removing this section. The paper does not focus on secondhand smoke, nor does it address the issue in detail. However, if you decide to retain it, I will respect that decision.

A: Thank you for your feedback. Our original intention was to provide a more comprehensive discussion of the harmful effects of smoking. However, after careful consideration of your comments, we have removed that section. We have chosen to focus more on the comorbidity of smoking and chronic diseases to emphasize the core content.

Reviewer #5:

1.For the methodology part, explain in detail about the inclusion and exclusion criteria for your participants

A: Thank you for your feedback. We have referred to numerous secondary analyses of public databases, particularly those using the CHARLS database. Such studies typically do not have fixed inclusion or exclusion criteria, as the research theme targets the entire elderly population in China. The key research variables are included in the analysis as long as there are no missing values.

2.For the discussion part, focus on comparing what your study has found versus what the existing literature has been shown. No need for over-explanation.

A: Thank you for your suggestion. We have also recognized this point and, after considering the feedback from other reviewers, revised, trimmed, and improved the discussion section. We have reduced excessive explanations and shifted the focus towards comparing our findings with other studies. The revised discussion emphasizes the key exploration of comorbidity patterns under different smoking statuses and their related factors. Line 237

These are my responses to all the reviewers' comments. I would like to express my sincere gratitude to the experts for their valuable input. Moving forward, I will approach my research with even greater diligence and enthusiasm. If there are any further comments or suggestions, I warmly welcome continued discussion and exchange.

---

## [Editor Report · Decision Letter 3]

27 Jan 2025

Analysis of chronic disease comorbidity patterns in middle-aged and elderly smokers in China: The China Health and Retirement Longitudinal Study

PONE-D-24-23479R3

Dear Dr. Zhang,

We’re pleased to inform you that your manuscript has been judged scientifically suitable for publication and will be formally accepted for publication once it meets all outstanding technical requirements.

Kind regards,

Saruna Ghimire

Academic Editor

PLOS ONE
---

## [Editor Report · Acceptance letter]

PONE-D-24-23479R3

PLOS ONE

Dear Dr. Zhang,

I'm pleased to inform you that your manuscript has been deemed suitable for publication in PLOS ONE. Congratulations! Your manuscript is now being handed over to our production team.

Kind regards,

on behalf of

Dr. Saruna Ghimire

Academic Editor

PLOS ONE